



# A 40-year global dataset of visible channel remote sensing reflectances and coccolithophore bloom occurrence derived from the Advanced Very High Resolution Radiometer catalogue.

Benjamin R. Loveday[1] and Timothy Smyth[1]

[1]Plymouth Marine Laboratory, Prospect Place, The Hoe, Plymouth, PL1 3DH, U.K.

**Correspondence:** Benjamin Loveday (blo@pml.ac.uk)

**Abstract.**

A consistently calibrated 40-year length dataset of visible channel remote sensing reflectance has been derived from the Advanced Very High Resolution Radiometer (AVHRR) sensor global time-series. The dataset uses as its source the Pathfinder Atmospheres - Extended (PATMOS-x) v5.3 Climate Data Record (CDR) for top-of-atmosphere (TOA) visible channel re-
5  flectances. This paper describes the theoretical basis for the atmospheric correction procedure and its subsequent implementation, including the necessary ancillary data files used and quality flags applied, in order to determine remote sensing reflectance. The resulting dataset is produced at daily, and archived at monthly, resolution, on a $0.1°$ x $0.1°$ grid at https://doi.pangaea.de/10.1594/PANGAEA.892175. The primary aim of deriving this dataset is to highlight regions of the global ocean affected by highly reflective blooms of the coccolithophorid *Emiliania Huxleyi* over the past 40 years.

## 1   Introduction

Remote sensing reflectance ($R_{rs}$), which has been listed as an Essential Climate Variable (ECV) by the Global Climate Observation System (GCOS), has been routinely monitored at the global scale by ocean colour satellites since the launch of the Sea viewing Wide Field of view Sensor (SeaWiFS) in September 1997. Prior to this, the proof-of-concept Coastal Zone Color
15  Scanner (CZCS) provided sporadic coverage for the period 1978 - 1986. Spectral $R_{rs}$ is a primary measurement of ocean colour satellites, and is used to determine higher level products such as Inherent Optical Properties (IOPs - Smyth et al. 2006), chlorophyll-a (O'Reilly et al., 1998) and Particulate Inorganic Carbon (PIC - Balch et al. 2005). $R_{rs}$ can also be used directly to detect brighter areas of the ocean caused by large blooms of the coccolithophorid *Emiliania Huxleyi*.

A subjective analysis, visually comparing global maps of coccolithophorid blooms during the CZCS era (Plate 1 from Brown
20  and Yoder 1994) and the first few years of the SeaWiFS mission (Figure 1 from Iglesias-Rodriguez et al. 2002), clearly shows large distributional changes in bloom occurrence between the two periods. However, the two analysis are separated by a decade where no ocean colour sensors were in operation. In the 1980s, Groom and Holligan (1987), published a coccolithophorid





bloom algorithm for use on visible channel Advanced Very-High Resolution Radiometer (AVHRR) data. The potential for using the AVHRR series of satellites, which spans the period between 1978 - present, as a means for bridging the observational gap between CZCS and SeaWiFS was seized upon by several studies (Morozov et al., 2013) with a particular emphasis on high latitude seas (Merico et al., 2003; Smyth et al., 2004). This built upon several studies in the 1980s and 1990s, before the

observational hiatus became an issue (Ackleson and Holligan, 1989; Matrai and Keller, 1993; Holligan et al., 1993; Garcia-Soto et al., 1995) and despite lower inherent reflectances in AVHRR channel 1 (0.580 - 0.680 $\mu$m) and lower detector gain rendering the sensor only 11% as sensitive to variation in coccolithophore reflectance as CZCS channel 3 (0.540 - 0.560 $\mu$m) (Groom and Holligan, 1987) and 3% as sensitive as SeaWiFS channel 5 (0.545 - 0.565 $\mu$m) (S. Groom *pers comm*).

Unfortunately, the lack of a consistent calibration between the different AVHRR sensors, and the difficulty in obtaining more

than individual orbit data has stymied a complete global analysis until now. Recently, PATMOS-X (Heidinger et al., 2014), a new satellite Climate Data Record (CDR) based on the continuous ~40 year global AVHRR visible channel record, has become available. Crucially, this has a consistent calibration across sensors and is geolocated on a 0.1° grid. Further information on the data set is available at https://cimss.ssec.wisc.edu/patmosx/.

In this paper we describe the derivation of a new dataset, which comprises a daily global $R_{rs}$ product and an associated

coccolithophorid bloom map. By using the consistent, well calibrated PATMOS-X base dataset this work will effectively double the current time period over which quantitative analyses of global $R_{rs}$ can be carried out from twenty to nearly forty years. It is over this order of observational time period that climatic shifts have been shown to be demonstrable (Henson et al., 2010).

## 2   Ingested data

The approach used here to derive $R_{rs}$ for AVHRR scenes is a modified version of that developed by Groom and Holligan (1987) and updated by Smyth et al. (2004). In the previous cases, calibrated radiances were derived from raw sensor counts, corrected for sensor degradation over time. However, as sensor degradation parameters are only available for AVHRR sensors on NOAA-7, 9, 11 and 14 (Rao and Chen, 1995, 1996), the approach is not applicable for analysis of long-term global signals. Here, rather than deriving the per-channel, top-of-atmosphere (TOA) reflectances using radiances calibrated

according to sensor-specific profiles, we use the cross-calibrated, degradation-corrected TOA reflectances extracted directly from version 5.3 of the Pathfinder Atmospheres - Extended (PATMOS-x) data set (Heidinger et al., 2014) (available at https://doi.org/10.7289/V56W982J, and subsequently referred to here as Px5.3).

Px5.3 is the first consistently gridded, climate quality data record of cross-calibrated AVHRR reflectances. It spans the period from 1979 to the present and contains between two and ten passes per day, dependent on the number of AVHRR instruments

operational on the TIROS-N, NOAA and MetOp platforms at the time (Figure 1). The $R_{rs}$ dataset derived from this record spans from 1979 to 2017, and includes the analysis of 62359 orbits. To calculate $R_{rs}$, we use the 0.63 $\mu$m (visible; channel-1) and 0.86 $\mu$m (near infra-red (NIR); channel-2). Channel-2 is predominantly used to correct for atmospheric aerosol effects, as the ocean is assumed to be dark in the NIR (e.g. $R_{rs}$=0).



To facilitate the atmospheric correction scheme, cloud-cover, water vapour and trace gas concentrations, winds, mean sea level pressure and sea surface temperature fields are extracted from the gridded, 6-hourly ERA-Interim products, provided by the European Centre for Medium Range Weather Forecasting (ECMWF) (available via https://www.ecmwf.int/en/forecasts/datasets/reanaly datasets/era-interim).

# 3 Method

## 3.1 Processing chain

Figure 2 presents a schematic diagram of the processing chain used to derive the $R_{rs}$ and the associated coccoliphophorid bloom map. The initial stages of the processor (QC1, atmospheric, $R_{rs}$, QC2 and scene output) are applied to each image in turn, prior to aggregation into a daily composite and a monthly climatology. Each stage of the processor is sequentially discussed below.

### 3.1.1 Initial quality control (QC1)

To prevent the calculation of erroneous $R_{rs}$ values, input reflectance data are masked according to a series of criteria based on measurement fidelity and consideration of the appropriate flags. The QC1 processor only retains reflectances where the following conditions are met:

- The cloud mask is equal to 0 (clear conditions).

- The glint mask is equal to 0 (no glint present).

- The land mask is not equal to 1 (permitting only ocean, coastal and inland water pixels).

- The "bad pixel" mask is equal to 0.

- The snow class mask is equal to 0 (no sea-ice).

- $0.0 \leq R_{TOA} \leq 1.0$ is satisfied for both channel 1 and channel 2.

- The sensor and solar zenith angles are finite and $< 90°$.

- The relative azimuth angle is finite.

Once these masking operations are complete, the QC1 processor passes the quality controlled TOA reflectances to the $R_{rs}$ processor, which awaits atmospheric inputs.



### 3.1.2 Atmospheric processor

Atmospheric data is required to calculate both the contribution of whitecaps to the ocean reflectance, and the gas absorbance transmission scaling factors for ozone, water vapour, $NO_2$ and $CO_2$. For each scene, the atmospheric processor bi-linearly interpolates the contemporaneous ERA-interim fields onto the Px5.3 grid, in both space and time. Wind speed, wind direction

and ozone concentration in Dobson Units (DU) are calculated from the gridded variables, and delivered to the $R_{rs}$ processor, along with transmission scaling factors.

### 3.1.3 $R_{rs}$ processor

The TOA reflectance signal contains both a water-leaving and, a much larger, atmospheric signal. To remove the atmospheric component of the signal, the contributions due to Rayleigh scattering, whitecaps and atmospheric absorbance must be quanti-

fied. In addition the effect of aerosols also needs to be accounted for. By assuming that the aerosol reflectance for channel-1 and channel-2 are equal (Stumpf and Pennock, 1989), and that $R_{rs}$ in channel-2 is zero, the effects of aerosols can be removed according to equation 1;

$$R_{rs_1} - R_{rs_2} = \frac{tw_0^s}{\pi \times tw_0^{pl}} \cdot \frac{(R_1 - R_2)}{\exp((0.5 \times \tau_0) \times pl)} \qquad (R_{rs_2} \to 0) \qquad (1)$$

where $pl$ is the atmospheric path-length, calculated according the satellite and solar zenith angles; $\tau_0$ is the Rayleigh optical

depth for channel-1 for a path-length of unity and; $tw_0^{pl}$ and $tw_0^s$ are the channel-1 transmission scaling factors for water vapour for $pl$ and in the sensor zenith direction (surface to sensor), respectively. $R_1$ and $R_2$ are the respective channel-1 and 2 TOA reflectances ($R^{TOA}$), corrected for Rayleigh scattering ($R^{Rayl}$), whitecaps ($R^{wcap}$) and atmospheric transmission.

Corrected reflectances are derived using equation 2;

$$R_n = \frac{1}{td_n^*} \left[ \frac{R_n^{TOA}}{tg_n^s \times tg_n^* \times td_n^s} - \frac{R_n^{Rayl}}{td_n^s} - R_n^{wcap} \right] \qquad (2)$$

where the $n$ subscript refers to the relevant channel (1 or 2). The transmission variables for atmospheric gases ($tg$), and the associated atmospheric scaling factors ($td$), are superscripted according to the sensor zenith ($s$) and solar zenith ($*$) directions. $R_{rs}$ values are not calculated where the Rayleigh reflectance calculation fails. The contribution of whitecaps is quantified using the method and look-up table described in Koepke 1984. Gas and water vapour absorption values are derived from Liang 2005 and Tanre et al. 1992.

### 3.1.4 Secondary Quality Control (QC2)

A secondary quality control protocol removes poor quality retrievals from the calculated $R_{rs}$ product, discarding pixels with negative $R_{rs}$ values. Whilst a rare occurrence, $R_{rs}$ pixels are also discarded where there is no acquisition time stamp as this



renders the calculated Rayleigh characteristics invalid. In this case, data within a two pixel mask of the erroneous point(s) are also discarded.

Periodically, low quality AVHRR data gives rise to patterns of erroneously high $R_{rs}$ values. Typically these aberrations effect a single pass, resulting in a poor quality 'stripe' across the R$_{rs}$ image. To remove this effect, each pass in the R$_{rs}$ product
is binned according to its integer hour of acquisition, which roughly corresponds to an individual pass (no specific pass number is available in the Px5.3 data). If a pass contains more than 5000 valid data points, and has a mean R$_{rs}$ value of higher than 0.001, the pass is considered to be of poor quality, and all data contained within it is discarded.

Over the South Atlantic, the Earth's Van Allen belt comes close to the planets surface. This 'South Atlantic Anomaly' causes excess radiation which can result in erroneous speckling in the AVHRR visible channel (Casadio and Arino, 2011). To remove
this effect each R$_{rs}$ product is subject to a filter, which removes pixels if they have a value that is greater than 5 times the maximum value of any of its neighbours. Coherent signals, associated with blooms, are unaffected. This process also removes single isolated pixels that are surrounded entirely by bad data.

When the solar zenith angle approaches $90°$ the number of counts in the visible channel drops substantially, degrading the quality of R$_{rs}$ estimate produced. To combat this effect, pixels where the number of counts in the 0.680 $\mu$m channel is less
than 10 are masked. Once the final R$_{rs}$ is calculated, it is written to an intermediary netCDF4 file during the "*Scene output*" stage. The pass-by-pass R$_{rs}$ product is not made available in this data set.

### 3.1.5 Compositing

For each day, the R$_{rs}$ products, calculated for each pixel on a pass-by-pass basis, are averaged into a single daily, global product. A daily product contains the average of anywhere between a two and ten passes, depending on the number of AVHRR
sensors in operation. Missing values are not included in the averaging process. In parallel, each pass is contributed to the "*Total aggregator*" stage, which calculates climatological monthly mean R$_{rs}$ values for each month, along with standard deviations and the number of observations available. Analogous statistics are also calculated for the total record. The "*Total aggregator*" stage can only be completed once all processed passes are available. Filtering for blooms cannot begin until the aggregator has concluded constructing the climatology. The final, unmasked, unfiltered R$_{rs}$ product is written into a daily composite netCDF4
file as "*remote_sensing_reflectance*", along with the original coordinate variables, as derived form the Px5.3 grid.

### 3.2 Filtering, masking and identifying blooms

In previous ocean colour based analyses, coccolithophorid bloom maps are produced as the binary classified output of supervised multi-spectral algorithm (e.g. Iglesias-Rodriguez et al. 2002; Brown and Yoder 1994). In this work, the availability of only one visible channel necessitates an alternative method, and the bloom map is instead produced through temporal filtering
of the R$_{rs}$ product, followed by selective masking to subsequently remove false positives.

Temporal filtering of the R$_{rs}$ product is performed through a comparison of each daily composite to the relevant monthly mean climatological R$_{rs}$ field (produced by the total aggregator stage). R$_{rs}$ signals are only classified as blooms where the per-pixel $R_{rs}$ value is greater than two standard deviations above the corresponding monthly mean value. The standard deviation





in this case is calculated from the monthly mean products across the entire archive. Pixels that do not match this criteria are assumed to contribute to the background, rather than bloom signal, and are therefore set to zero. The filtered bloom product, written into the daily netCDF4 file as *filtered_remote_sensing_product* is then subject to further quality controls in the "*Masking*" stage, as described below.

High $R_{rs}$ values, while potentially indicative of coccolithophore blooms, can also occur in regions that are subject to high concentrations of suspended sediment (e.g. estuaries), or where shallow bathymetry and clear water coincide (e.g. shelf regions in oligotrophic areas). To remove these, and other false positives, the final bloom product is derived from the filtered bloom product by subjecting the latter to a number of screening processes, as detailed below.

Firstly, to remove the effects of land contamination, the bloom map is set to zero in all points within 3 pixels of the land mask.
Secondly, following Iglesias-Rodriguez et al. 2002, the bloom map is set to zero in areas between 47°N and 47°S where the bathymetry is shallower than 100 m. This removes false positives associated with the sea floor, an effect that is most noticeable in the Caribbean and Arafura Seas. Thirdly, whilst flagged sea-ice has been explicitly removed from the Px5.3 data (see section 3.1.1), this does not comprehensively remove ice effects. As a result of missed flagging, and of glacial (Broerse et al., 2003) and river run off, sporadic high $R_{rs}$ values that are not indicative of blooms still occur at high-latitudes. To correct for this,
bloom map pixels are set to zero where $R_{rs} \geq 0.05$ (a value far above that which we would expect in water types associated with coccolithophorid blooms). Furthermore, the $R_{rs}$ product is screened using sea surface temperature (SST) data obtained from contemporaneous ERA-interim fields, and $R_{rs}$ is set to zero in pixels where SST < 0°C in the northern hemisphere, a value at which the coccolithophorid growth rate drops to near zero, even for cold-water strains such as *E. huxleyi* (Buitenhuis et al., 2008). Finally, bloom map pixels are set zero where the total aggregated mean value $R_{rs}$ is greater than 0.0005, removing
the effects of consistent river outflows (e.g. along the Amazonian coast, and in the Yellow Sea).

The final product suite is annotated with relevant metadata to ensure CF1.8 compliance, completing the processing. The contents of the data file are described fully in the following section.

## 4    Data Provenance and Structure

The complete finalised data set consists of 13932 daily files, beginning on January $1^{st}$ 1979 and ending on December $31^{st}$
2017. Table 1 describes periods where data is missing, either due to a lack of available AVHRR data in the Px5.3 archive, or lack of viable data for $R_{rs}$ processing. Completely empty scenes are not in included in the archive.

The products are provided at 0.1° resolution (consistent with the original Px5.3 grid). Each data file, contains the variables listed in table 2.

Responsibility for maintaining the dataset lies with Plymouth Marine Laboratory, the provenance authority for the final
output (Figure 2). The data set will be updated periodically, but no specific update schedule is set. The initial release version is v1.0. Minor version updates to bring the archive up to date will increment the decimal value. Major updates, in the case of changes to processing will increment the integer value.



**Table 1.** Inventory of missing $R_{rs}$ products in the processed archive due to missing or unviable AVHRR data.

| Year | missing days |
|---|---|
| 1979 | 21/02 to 25/02, 03/04, 18/05, 14/07, 16/07, 28/07, 01/10 to 07/10, 02/11 to 07/11, 10/11, 15/11, 18/11, 30/11, 12/12 |
| 1980 | 20/01 to 30/06, 03/07, 05/07, 06/07, 09/07, 11/07 to 19/07, 07/08, 11/08, 12/08, 13/08, 14/08, 01/09, 07/09, 09/09, 12/09, 04/10, 22/10, 23/10, 07/12, 12/12 to 18/12, 25/12, 27/12, 29/12, 30/12 |
| 1981 | 08/01, 21/01, 25/03, 03/04, 09/05 to 11/05, 16/06, 25/06, 27/06, 03/07, 01/08 to 04/08, 14/08, 15/08, 17/08, 20/08 to 23/08 |
| 1982 | 24/04, 28/04, 03/05, 04/05, 09/05, 28/05 to 31/05, 01/06, 03/06, 25/09, 16/09, 29/09 |
| 1983 | 06/08, 24/08 |
| 1984 | 14/01, 15/01, 27/01, 20/02 to 22/02, 23/03, 24/03, 10/04, 16/04, 17/06, 29/07, 06/12, 07/12 |
| 1985 | 02/02 to 24/02, 11/03 |
| 1986 | 14/03, 15/03 |
| 1990 | 06/02 |

**Table 2.** Fields present in the available data

| Variable name | Quantity | Units | Dimensions |
|---|---|---|---|
| time | time | seconds since 1970-01-01 00:00:00 | time[1] |
| latitude | latitude | degrees North (-89.948 to 89.948) | latitude[1800] |
| longitude | longitude | degrees East (-179.945 to 179.945) | longitude[3600] |
| remote_sensing_reflectance | $R_{rs}$ | sr$^{-1}$ | time x latitude x longitude |
| filtered_remote_sensing_reflectance | filtered $R_{rs}$ bloom product | sr$^{-1}$ | time x latitude x longitude |

Due to the size of the entire daily-resolution record (> 60 Gb in total), the data is archived one monthly time-base, with monthly mean and maximum fields available as separate files. The dataset is stored in the PANGAEA archive, and has the following digital object identifier: https://doi.pangaea.de/10.1594/PANGAEA.892175.



## 5   Bloom product validity

### 5.1   Regional comparisons

Figure 3 shows a comparison between four coccolithophorid blooms detected by ocean colour sensors and the corresponding blooms in the *filtered_remote_sensing_reflectance* product. In all four cases where bright blooms are detected in the ocean colour sensor observations (a) SeaWiFS; c) MERIS; e) and g) MODIS) there are spatially corresponding bright patches in the AVHRR imagery. In the MERIS and MODIS cases the AVHRR imagery is from the same day (i.e. on a single overpass basis). In the SeaWiFS case, a 3-day AVHRR composite mean is used, due to differences in cloud cover at the various acquisition times, lowering the intensity of the visible bloom, but preserving the spatial coverage of the scene.

There is also evidence from *in situ* data in the English Channel case (Figure 3 a) and b)) that this was indeed a bloom of *Emiliania Huxleyi* from cell counts and in-water radiometry (Smyth et al., 2002). The northern North Sea feature centred on 56°N 1.5°E in Figure 3 a) is possibly the remnants of a bloom which was the subject of an intensive field campaign during June 1999 (Burkill et al., 2002). Similarly blooms have been documented in the literature in the Barents Sea (Figure 3 e) and f)) which are comparable in timing and extent with some limited *in situ* samples (Smyth et al., 2004); Merico et al. (2003) report on blooms in the Bering Sea of similar timing and extent to those shown in Figure 3 g) and h).

It is notable that the cloud (and ice) masking algorithms for the ocean colour and AVHRR sensors are in close agreement for the individual scenes shown in Figure 3 c) and d); e) and f) and; g) and h). The discrepancy in the cloud flagging for the SeaWiFS case occurs as a result of the 3-day composite discussed previously.

### 5.2   Global signals

Figure 4 shows the global spatial coverage of the data set, presented as decadal means of the filtered $R_{rs}$ bloom product for four decadal periods. Comparing figure 4 panel a) with the CZCS era (1978-1986) coccolithophorid bloom map produced by Brown and Yoder (1994) suggests that the bloom signals in the North Atlantic, North Sea, Norwegian Sea, and Argentinian coast are well captured. The presence of blooms in the Black Sea and sporadically throughout the Mediterranean is also consistent between the two. However, due to the coastal masking, the high signals around Newfoundland are not captured here. In addition, no signals are detected in the Arafura Sea and within the Indonesian archipelago, as these areas have been specifically masked due to shallow bathymetry. The signals along the coast of Greenland and in the Southern Ocean are stronger than anticipated.

Mean values for the 1990-1999 period suggest the presence of coccolithophorid blooms in the North Atlantic, Norwegian Sea, Baltic Sea, Bering Sea and Southern Ocean, are consistent with previous findings of Iglesias-Rodriguez et al. (2002). Similarly, the $R_{rs}$ based bloom product correctly attributes signals to the Benguela upwelling and in the North West Pacific Ocean. While they do not cover identical periods to the record shown here, increased $R_{rs}$ in the Black Sea, Norwegian Sea and Baltic between the Brown and Yoder (1994) and Iglesias-Rodriguez et al. (2002) studies is well replicated. A reduction of $R_{rs}$ along the coast of Argentina also appears to be appropriately captured.



## 6   Limitations

### 6.1   Radiometric sensitivity and grid resolution

The differences between the bloom extent and intensity observed by the ocean colour and AVHRR sensors in figure 3 can, in part, be attributed to a combination of lower spatial resolution and radiometric sensitivity of the AVHRR sensor. The typical
spatial resolution of ocean colour sensors is typically between 300 m and 1.1 km, whereas the Px5.3 product is 0.1°. This equates to a two orders of magnitude of coarsening, and will have a particularly pronounced effect where strong spatial radiometric heterogeneities exist within blooms (see e.g. Smyth et al. (2002)), resulting in lower overall bloom reflectances. This coupled with the lower radiometric sensitivity of AVHRR (3%) will exacerbate the overall "dimming" of the bloom. By similar logic, it is unlikely that non-coccolithophorid blooms will be detectable using this approach.

### 6.2   Atmospheric correction

The availability of only a single visible channel, greatly reduces the ability of AVHRR to spectrally determine in water constituents. Consequently, coccolithophorid blooms are predominantly identified through the removal of likely false positives (e.g river plumes, coastal influences, bathymetry and sea-ice). While coccolithophorid blooms are known to occur across the Eastern (Malinverno et al., 2003) and North Western (Oviedo et al., 2015) and Western Mediterranean (Ignatiades et al., 2009),
as shown in the bloom map of Iglesias-Rodriguez et al. (2002), they do not do so to the extent that they are expressed in the $R_{rs}$ product derived here (figure 4, panel b). We partially attribute the high $R_{rs}$ values to the sporadic presence of Saharan dust. This requires specific atmospheric correction methods (Moulin et al., 2001) that are not implemented here, and would require ancillary dust information that is of limited availability on the time scale covered by this data set.

## 7   Conclusions

We have derived a consistently calibrated 40-year length dataset of visible channel remote sensing reflectance from the Advanced Very High Resolution Radiometer (AVHRR) sensor global time-series. We have shown how this global dataset was derived from top-of-atmosphere (TOA) visible channel reflectances including how the data was quality controlled, atmospherically corrected and aggregated over daily, monthly and decadal time-periods. We have shown the application of this new dataset to the detection of marine phytoplankton and compared these to existing regional and global imagery and estimates
from different satellite sensors and *in situ* data. We have effectively extended the time-period over which the detection of coccolithophorids is possible on the global scale by an additional 20 years, thereby making possible further analyses of climatic shifts in species distribution.





## 8 Code and data availability

The data set is registered and archived with a digital object identifier at PANGAEA. It is made available for use with the following reference: https://doi.pangaea.de/10.1594/PANGAEA.892175. The code used to generate this data is available via Git on request to the corresponding author.

5  *Author contributions.* Ben Loveday and Tim Smyth contributed equally to the writing of the manuscript and data quality control. Ben Loveday is responsible for the data-set processing architecture and comparative analysis with ocean colour. Tim Smyth was the originator of the concept.

*Competing interests.* The authors declare that they have no competing interests.

*Acknowledgements.* This work was funded by the 2017/2018 Plymouth Marine Laboratory Research Programme. The authors would like to
10  thank Dr. Hayley Evers-King for constructive criticism of the manuscript.



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



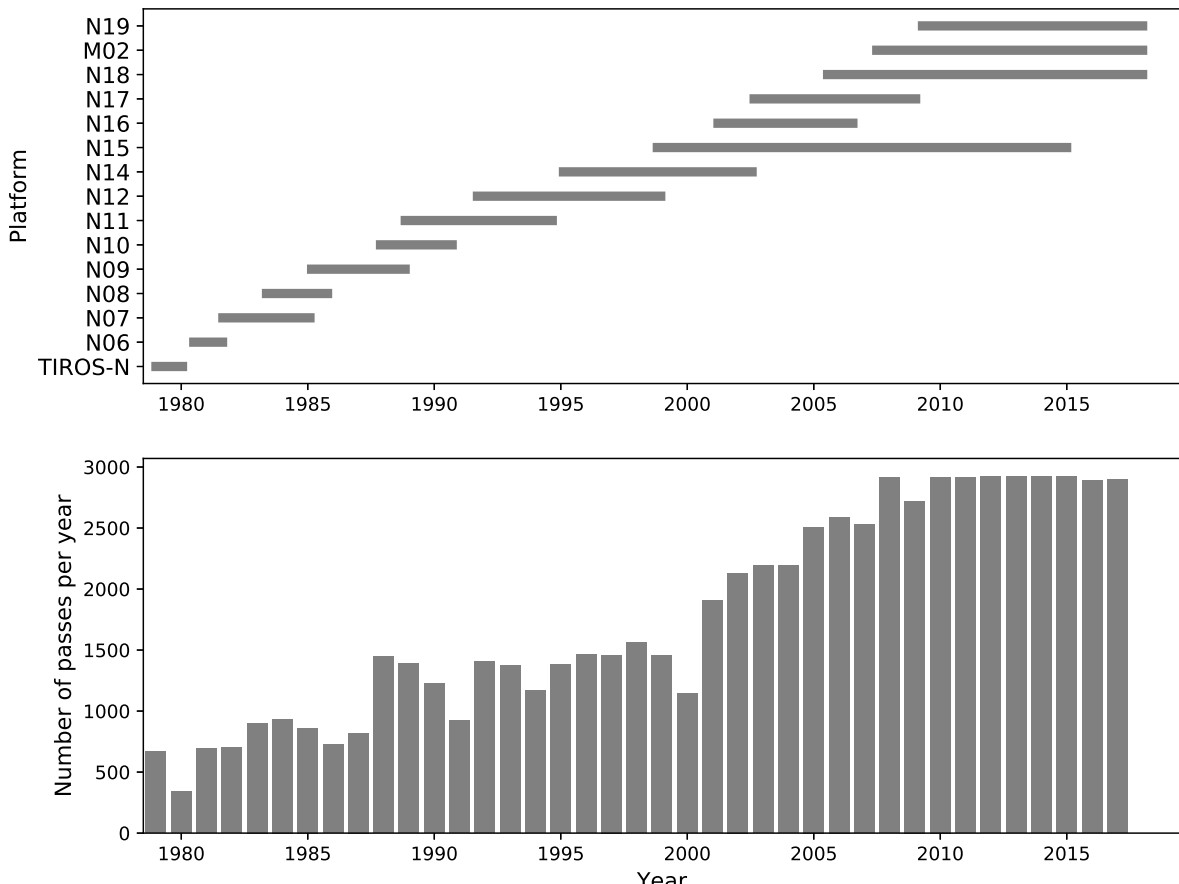

**Figure 1.** PATMOS-x v5.3 (Px5.3) data density. Top panel: length of the individual AVHRR missions where TIROS-N (Television Infra-Red Observation Satellite) operated by NASA, N refers to NOAA operated missions, and M to MetOp missions, operated by EUMETSAT. Bottom panel: number of satellite orbits per year which comprise the Px5.3 dataset.





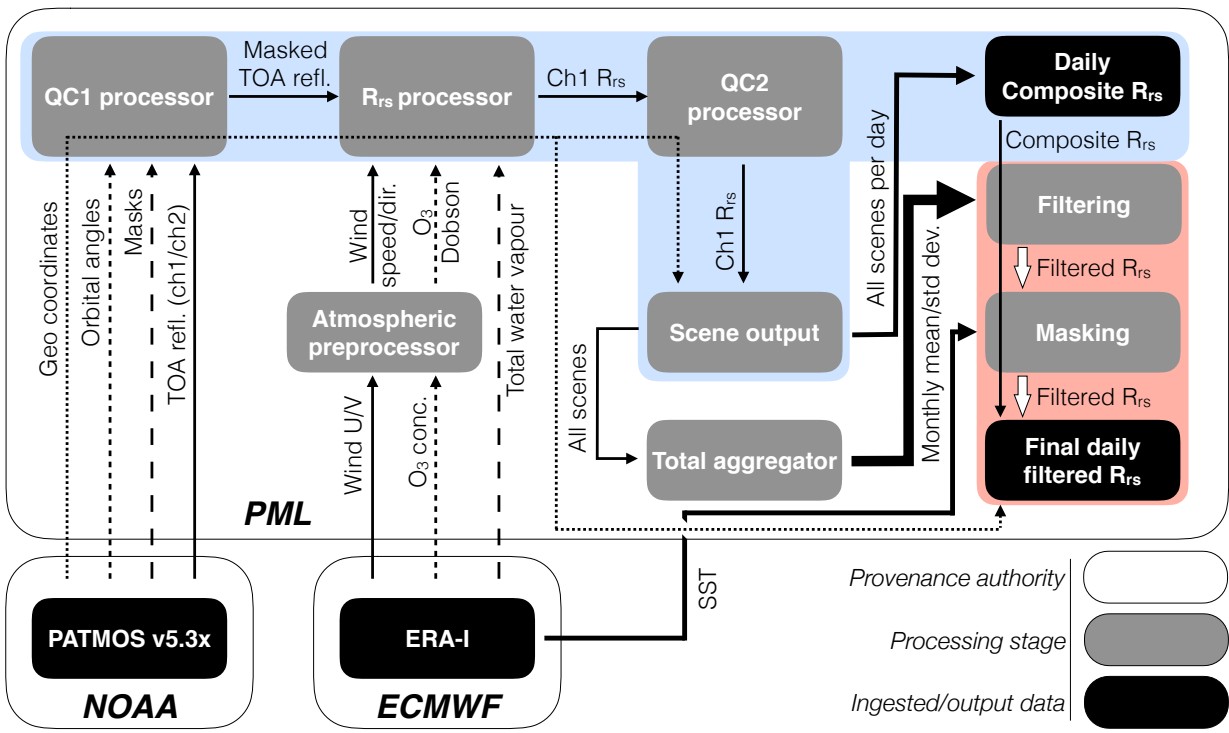

**Figure 2.** Schematic diagram showing the different stages of the remote sensing reflectance ($R_{rs}$) processing chain. The blue-shaded region generated the unfiltered $R_{rs}$ product, the red-shaded region subsequently generates the filtered $R_{rs}$ product. PML, NOAA and ECMWF refer to Plymouth Marine Laboratory, the National Oceanic and Atmospheric Administration and the European Centre for Medium Range Weather Forecasting, respectively.





**Figure 3.** Examples of ocean colour derived red-green-blue (RGB) images of coccolithophore blooms matched to their filtered bloom product counterparts. Panels are as follows a,c,e,g) Level 2 RGB images for the North Sea and English Channel (SeaWIFS; 30th July 1999), North Atlantic and Irish Sea (MERIS; May 23rd 2010), Barents Sea (MODIS; 17th August 2011) and Bering Sea (MODIS; 4th September 2014). b,d,f,h) Matching, contemporaneous filtered bloom product composite for each location and date. Dark gray indicates land and light gray indicates cloud, throughout. For bloom products, dark blue indicates that no bloom is present, lighter cyan colours indicate that a bloom is present.



**Figure 4.** Mean values for the filtered remote sensing reflectance ($R_{rs}$) bloom product by decade for the a) 1980 to 1989, b) 1990 to 1999, c) 2000 to 2009 and d) the abbreviated 2010 to 2017 period.