# Peer review of "A 40-year global dataset of visible channel remote sensing reflectances and coccolithophore bloom occurrence derived from the Advanced Very High Resolution Radiometer catalogue."

_Earth System Science Data, 2018_

## Referee Comment (RC1) · Anonymous Referee #1 · 2 Aug 2018

General appraisal

Loveday and Smyth have generated a 40-year long dataset of coccolithophore blooms occurence over the global ocean, from observations of the Advanced Very High Resolution Radiometer (AVHRR) visible bands.

This is a very timely effort, inasmuch as generating long-term, consistently calibrated satellite time series is absolutely needed for studies about how global environment

changes affect marine ecosystems.

The paper is overall well written, concise, and including the appropriate level of details. Illustrations are of good quality. So, overall, an excellent paper.

My only reservation would be about the attribution of the high reflectance signal to coccolithophore blooms. The authors themselves recognise that they cannot always ascribe the reflectance anomalies to the presence of such blooms (their discussion on limitations, page 9).

I do not think that referring simply to highly scattering waters instead of coccolithophore blooms would undermine the paper value. Any subsequent user of the data set can bring his own interpretation of what these high-scattering waters are, depending on the area and season under investigation (even if, admittedly, they are probably most of the time caused by the presence of coccoliths). It might even prevent potential users to actually negatively comment on this data set because they would have found a clear example of such high-scattering waters not being due to the presence coccoliths, whereas the data set "claims" that they are.

Some minor comments

- Lines 20-27 page 2: this whole paragraph is rather unclear to me. Not sure what has been done at the end. Maybe this could be expanded a bit.

- Line 21 page 3: could the authors have used a lower threshold, to account for the fact that observations will anyway be hardly exploitable for large sun zenith angles, roughly above $70°$?

- Eq. (1): I suspect a "-" sign is missing before the 0.5 in the denominator.

- Eqs. (1) and (2): maybe Eq. (2) should actually come first, because that is where R is defined, and then the computation of Rrs from R would be Eq. (2). My other comment here is that these equations could maybe be split into a few more equations, to more clearly show the logic.

- Line 4 page 5: I guess it is "affect", not "effect"

- Line 7 page 5: "are discarded"

- Line 19, page 5: "between two and ten"

- Line 20, page 5: I do not understand the point here. I do not see how missing values could anyway be included in an average. Maybe rephrase.

- Line 19 page 6: "set to zero"

- Line 25 page 6: "are missing"

- Line 26, page 6: "not included"

- First line of the paragraph page 7: "archived on a "

- Line 20 page 8: maybe this CZCS map could be included here to facilitate comparison
* * *

---

## Referee Comment (RC2) · G. NEUKERMANS (Referee) · 10 Aug 2018

This work provides a 40-year long global record of coccolithophore (Emiliania huxleyi species) bloom observations from a meteorological satellite (AVHRR). The dataset potentially provides by far the longest record of global observations of blooms of a single phytoplankton species, Emiliania huxleyi, available to date, and thereby provides a time series long enough to detect changes due to anthropogenic climate change. The work

takes advantage from recent efforts by the meteorological community who provided a consistently calibrated record of top-of-atmosphere reflectances across all AVHRR satellite missions. Retrieval of a marine signal (remote sensing reflectance) from top-of-atmosphere reflectance, however, requires a careful removal of atmospheric effects (atmospheric correction). My major criticisms are that (1) the atmospheric correction procedure, which is the most important processing step, is not sufficiently well documented in the present paper and may need to be revised, (2) the resulting remote sensing reflectance dataset lacks validation, (3) the detection limit and uncertainties of the dataset are not quantified, which are problematic for potential users of the data. Below are my suggestions for improvement of the work on a point-by-point basis and a list of relatively minor comments and suggestions.

(1) Atmospheric correction:

I would suggest that section 3.1.3 be expanded and logically restructured to include: (1) equations documenting the appropriate corrections in each subsequent processing step, starting with a breakdown of the different component reflectances and transmittances of the top-of-atmosphere reflectance. (2) A clear mention of all assumptions made and how they impact the retrieved remote sensing reflectance. E.g., How does the assumption of equal aerosol reflectances in the NIR and the VIS affect Rrs(VIS)? (3) The relationship between transmission scaling factors and one-or-two way transmisttance values. (4) A revision of Equation 1 (I believe you are missing a minus sign in the denominator). (5) A documentation of the radiative transfer code used to compute Rayleigh reflectance, atmospheric transmittances etc. (6) the relationships used between gas concentrations and gas transmittances. (7) Did you correct for $CO_2$ absorption (it does not appear in Figure 2)? (8) an explanation as to how and why the approach differs the one documented in Smyth et al. (2004). (9) Did you correct for aerosol transmittance (not mentioned)? (10) Is there a sunglint correction? (11) Briefly describe the whitecap correction you used.

(2) Rrs(VIS) validation :

Even though a qualitative validation is presented in Figure 3 through a visual compari-
son of four scenes from ocean colour satellites and corresponding AVHRR reflectance
products, this work would clearly benefit from a more quantitative assessment. A cross-
validation between Rrs(VIS) from appropriate ocean colour satellite bands and Rrs(VIS
AVHRR) is probably the best way to validate the dataset produced. Additionally you
could compare bloom extent from AVHRR with the coccolithophore bloom mask of
Brown and Yoder applied to ocean colour satellites. There is about 20 years of over-
lapping satellite observations between AVHRR and ocean colour satellites; this should
be plenty to make a quantitative assessment of the quality of your dataset. Further,
many older papers provided AVHRR satellite data of Emiliania blooms at the original
resolution of the sensor, 1km. A comparison with those data would allow you to esti-
mate just how problematic the degradation of the spatial resolution by PATMOS is.

(3) Detection limit and uncertainties of the retrieved Rrs and bloom product

Just how reflective does a bloom need to be in order to be picked up by AVHRR satel-
lites and how does the detection limit differ among missions? Can you give typical con-
centrations of coccolith(ophore)s corresponding to those reflectance detection limits?
These questions are relevant for biologists, ecologists, and biogeochemists interested
in coccolithophore blooms. Even if the detection limit would vary across sensors and
with illumination and viewing geometry, it would be necessary to document this both in
the paper as well as an additional variable in the dataset.

Another major comment relates to the poor spatial and temporal resolution of the
remote sensing reflectance dataset which is monthly at 0.1°x0.1°. As mentioned
in the paper this is a spatial coarsening by two orders of magnitude from the
original resolution of AVHRR satellite data. Such a coarsening apparently results
in a "dimming" of coccolithophore bloom intensity, which is even further exacer-
bated by making monthly composites. Uz et al. (2014) [Monitoring a Sentinel
Species from Satellites: Detecting Emiliania huxleyi in 25 Years of AVHRR Imagery,
https://link.springer.com/chapter/10.1007/978-94-007-5872-8_18 ] also used a consistently calibrated dataset of AVHRR data processed with CLAVR-X at 0.25°x0.25° spatial resolution. Why is this dataset not used instead? It gives much better spatial resolution, and will partly solve your problem of "bloom dimming". Further, as your work is very similar to the work of Uz et al it is an important reference currently missing in your paper. Another (easier) way to mediate the problem of bloom dimming is by increasing the temporal resolution from monthly to weekly. I think weekly resolution data would also allow users to investigate bloom phenology which is not possible with monthly data.

My last major comment relates to the temporal filtering applied to detect blooms (P5L32-33). I think you are missing out on quite a bit of bloom area by considering as blooms only those observations which have Rrs more than two standard deviations above the corresponding monthly climatological mean. If annual blooms occur in an area in the same month almost every year and last a couple of weeks (which they typically do!) this will give rise to high climatological mean values. Your filtering will miss those blooms as they are considered "background".

Minor comments:

(1) P1L9: Rrs detection limit needs to be quantified in the abstract (2) P1L17-18: You should explain why Emiliania huxleyi blooms result in bright waters (3) P1L22: A reference to the paper by Winter et al (2014) [Poleward expansion of the coccolithophore Emiliania huxleyi, https://academic.oup.com/plankt/article/36/2/316/1500265 ] would be appropriate (4) P2L10: "Until now" but see Uz et al. (5) P2L28: what about the dataset used in Uz et al? (6) P2L32: worth mentioning the spectral width of Channel 2 (7) P5L14-16: Why don't you just mask high solar zenith angles? As it is now you are loosing good quality clear water, clear atmosphere data (8) P6L11: I would recommend applying a bathymetry mask poleward of 47° as well as false positives are also found on the shallow shelves of the Bering (60°N) and Barents (70°N)Seas. (9) P8L26: do you have an explanation for this? (10) P8L33: reference missing (11) P8 Section 5.2: Your maps in Figure 4 are further consistent

with other studies documenting poleward expansion of Emiliania huxleyi blooms by Winter et al. (2014) and Neukermans et al. (2018) [Increased intrusion of warming Atlantic water leads to rapid expansion of temperate phytoplankton in the Arctic, https://onlinelibrary.wiley.com/doi/abs/10.1111/gcb.14075 ]. Both references are missing in the paper. (12) P9L3-4: typical is used twice (13) P9L9: "By similar logic" – strange comment to make as blooms of non-calcifying phytoplankton are typically detected based on absorption features, not backscattering.

---

## Author Comment (AC1) · 15 Oct 2018

—————————
**Response to RC1**
—————————

**Reviewer comments in black**
**Author responses in blue**
**Manuscript changes in green**

*Anonymous Referee 1*

**General appraisal:**

Loveday and Smyth have generated a 40-year long dataset of coccolithophore blooms occurence over the global ocean, from observations of the Advanced Very High Resolution Radiometer (AVHRR) visible bands.

This is a very timely effort, inasmuch as generating long-term, consistently calibrated satellite time series is absolutely needed for studies about how global environment changes affect marine ecosystems.

The paper is overall well written, concise, and including the appropriate level of details. Illustrations are of good quality. So, overall, an excellent paper.

**Major comments:**

My only reservation would be about the attribution of the high reflectance signal to coccolithophore blooms. The authors themselves recognise that they cannot always ascribe the reflectance anomalies to the presence of such blooms (their discussion on limitations, page 9).

I do not think that referring simply to highly scattering waters instead of coccolithophore

blooms would undermine the paper value. Any subsequent user of the data set can bring his own interpretation of what these high-scattering waters are, depending on the area and season under investigation (even if, admittedly, they are probably most of the time caused by the presence of coccoliths). It might even prevent potential users to actually negatively comment on this data set because they would have found a clear example of such high-scattering waters not being due to the presence coccoliths, whereas the data set "claims" that they are.

The specific goals of this paper are two fold:

1. To produce an atmospherically corrected visible channel Rrs product from the PATMOS-X processed AVHRR climate data record of top of atmosphere reflectances; and

2. To introduce a filtering procedure that isolates the occurrence of coccolithophorid blooms in this record.

Consequently, we would agree with the reviewer with regard to the attribution of the high-reflectance signal, had the filtered product not also been provided. Our feeling is that recasting the paper as isolating 'high-scattering waters' somewhat undermines the work taken to isolate the coccolithophorid blooms through the filtering procedure, and reduces the appeal of the paper.

Previous work by Uz et al., 2013 (Monitoring a sentinel species from satellites: detecting Emiliania huxleyi in 25 years of AVHRR imagery), Smyth et al. 2004 (Time series of coccolithophore activity in the Barents Sea, from twenty years of satellite imagery) would not satisfy the reviewers suggestion, despite applying broadly similar, if not less stringent, approaches.

The point regarding the discussion of false positives in the limitations is well taken. However, extensive care has been taken to remove erroneous data, bathymetric and
high-sediment load effects, low signal contamination at high solar zenith angle and all semi-permanent or 'bright background' signals. Over time, it may prove that the discussion of the limitations of the filtered product may not be exhaustive. However, we maintain that the vast majority of filtered detections would be due to coccolithophorid blooms (as the reviewer themselves kindly notes).

**Some minor comments:**

Lines 20-27 page 2: this whole paragraph is rather unclear to me. Not sure what has been done at the end. Maybe this could be expanded a bit.

This paragraph has been entirely re-written to improve clarity, add further information and make better references to how our approach differs from previous work.

Previous, efforts to derive visible channel $R_{rs}$ from the AVHRR catalogue (e.g. Groom and Holligan (1987) and Smyth et al. (2004)) typically use the raw, instrument counts as a starting point to calculate per-channel TOA radiance. In order to apply this approach across the lifetime of a single AVHRR sensor, the radiance must be calibrated according to the sensor degradation parameters. However, as sensor degradation parameters are only available for AVHRR sensors on NOAA-7, 9, 11 and 14 (Rao and Chen , 1995, 1996), the approach is not applicable for analysis of long-term global signals. Consequently, here we adopt a modified version of the approach used by Groom and Holligan (1987), and updated by Smyth et al. (2004), which uses the TOA reflectances as a starting point for the atmospheric correction procedure. The approach is fully detailed in section 3.1.3.

Per channel TOA reflectances are extracted directly from version 5.3 of the Pathfinder Atmospheres - Extended (PATMOS-x) data set (Heidinger et al., 2014) (available at https://doi.org/10.7289/V56W982J) and subsequently referred to here as Px5.3). Px5.3 reflectances are inter-calibrated across AVHRR sensors, and are corrected for sensor degradation throughout. Px5.3 is the first consistently gridded, climate quality data

record of cross-calibrated AVHRR reflectances. It spans the period from 1979 to the present and contains between two and ten passes per day, dependent on the number of AVHRR instruments operational on the TIROS-N, NOAA and MetOp platforms at the time (Figure 1). The $R_{rs}$ dataset derived from this record spans from 1979 to 2017, and includes the analysis of 62359 orbits. To calculate $R_{rs}$, we use the 0.63 $\mu$m (visible; channel-1) and 0.86 $\mu$m (near infra-red (NIR); channel-2). Channel-2 is predominantly used to correct for atmospheric aerosol effects, as the ocean is assumed to be dark in the NIR (e.g. $R_{rs}$=0).

Line 21 page 3: could the authors have used a lower threshold, to account for the fact that observations will anyway be hardly exploitable for large sun zenith angles, roughly above 70$^o$?

The implementation of a solar zenith angle cut-off is implemented solely as a first pass quality control in the initial data ingestion phase and removes erroneously reported angles throughout the data set. Implementation of a stricter filter (e.g. 70$^o$), would result in a substantial loss of data coverage, especially at high latitudes, where blooms are common. Later filtering of the data in the quality control 2 stage removes points where the signal becomes unusably low due to the radiometric sensitivity of AVHRR.

Eq. (1): I suspect a "-" sign is missing before the 0.5 in the denominator.

The reviewer is correct. This has been updated.

Eqs. (1) and (2): maybe Eq. (2) should actually come first, because that is where R is defined, and then the computation of Rrs from R would be Eq. (2). My other comment here is that these equations could maybe be split into a few more equations, to more clearly show the logic.

We agree with both reviewers that the previous presentation of the equations used was not overly logical, and significantly more information on the approaches used is required.

[Figure]

The order in which equations 1 and 2 were presented has been reversed, and more information is given on the treatment of the data as a whole. As part of this process, the equations which are used to correct various atmospheric contributions are now presented separately. These changes have resulted in extensive changes to section 3.1.3, which are too extensive to replicate here.

Line 4 page 5: I guess it is "affect", not "effect"

The reviewer is correct. This has been updated.

Line 7 page 5: "are discarded"

Corrected

Line 19, page 5: "between two and ten"

Corrected

Line 20, page 5: I do not understand the point here. I do not see how missing values could anyway be included in an average. Maybe rephrase.

The phrase: "Missing values are not included in the averaging process", has been re-written to "Values recorded as missing or filled values in the individual netCDF4 products are masked, and are therefore not included in the averaging process"

Line 19 page 6: "set to zero"

Corrected

Line 25 page 6: "are missing"

Corrected

Line 26, page 6: "not included"

Corrected

First line of the paragraph page 7: "archived on a "

Corrected

Line 20 page 8: maybe this CZCS map could be included here to facilitate comparison.

Unfortunately, it has not proven possible to obtain a copy of this panel in sufficiently high resolution to be suitable for inclusion in this publication.

––––––––––––––––––––––––

---

## Author Comment (AC2) · 15 Oct 2018

—————————
**Response to RC2**
—————————

**Reviewer comments in black**
**Author responses in blue**
**Manuscript changes in green**

*Anonymous Referee 1*

**General appraisal:**

This work provides a 40-year long global record of coccolithophore (Emiliania huxleyi species) bloom observations from a meteorological satellite (AVHRR). The dataset potentially provides by far the longest record of global observations of blooms of a single phytoplankton species, Emiliania huxleyi, available to date, and thereby provides a time series long enough to detect changes due to anthropogenic climate change. The work takes advantage from recent efforts by the meteorological community who provided a consistently calibrated record of top-of-atmosphere reflectances across all AVHRR satellite missions. Retrieval of a marine signal (remote sensing reflectance) from top-of-atmosphere reflectance, however, requires a careful removal of atmospheric effects (atmospheric correction). My major criticisms are that (1) the atmospheric correction procedure, which is the most important processing step, is not sufficiently well documented in the present paper and may need to be revised, (2) the resulting remote sensing reflectance dataset lacks validation, (3) the detection limit and uncertainties of the dataset are not quantified, which are problematic for potential users of the data. Below are my suggestions for improvement of the work on a point-by-point basis and a list of relatively minor comments and suggestions.
**Major comments:**

**Atmospheric correction:**

I would suggest that section 3.1.3 be expanded and logically restructured to include:

(1a) Equations documenting the appropriate corrections in each subsequent processing step, starting with a breakdown of the different component reflectances and transmittances of the top-of-atmosphere reflectance.

This section has now been extensively re-written, and expanded according to the reviewer's wishes, beginning with what we hope is now a clear treatment of each reflectance component in turn.

Please see extensive changes to section 3.1.3, page 4, line 13 onwards.

(1b) A clear mention of all assumptions made and how they impact the retrieved remote sensing reflectance. E.g., How does the assumption of equal aerosol reflectances in the NIR and the VIS affect Rrs(VIS)?

Section 3.1.3 now includes text detailing the assumptions made during the atmospheric correction procedure. However, while modern ocean colour satellites have 2 bands in the NIR to allow for correction of the aerosol component through interpolation, AVHRR does not benefit from this capability. Consequently, it is not possible for us to assess the effect that the equal aerosol assumption has on Rrs(VIS) as we cannot quantify the aerosol component independently. This lack of available data was a main driver in adopting the channel balance approach, as used in previous studies (Groom and Holligan, 1987; Smyth et al., 2004, Uz et al., 2013).

(1c) The relationship between transmission scaling factors and one-or-two way transmittance values.

This is now explicitly discussed in the text at the end of section 3.1.3.

Please see manuscript page 6 line 5-16.

(1d) A revision of Equation 1 (I believe you are missing a minus sign in the denominator).

The reviewer is correct. This has been updated.

In addition, equation 7 (previously equation 1) was presented incorrectly. In our implementation, water vapour transmission is encapsulated within the per-channel gas transfer calculations, and not as a final scale factor. The corrected equation accurately mirrors the atmospheric correction used here.

(1e) A documentation of the radiative transfer code used to compute Rayleigh reflectance, atmospheric transmittances etc.

We are not clear on what the reviewer requires here. The full code is extensive, and we have noted in the manuscript that it is available on git should users require it. However, fully documentation of the code in the manuscript would be extremely cumbersome for the reader. The manuscript now provides much more depth as to what processes have been performed, and we would hope that this satisfies their requirements under this point.

Please see section 3.1.3, page 5, lines 8-16.

(1f) The relationships used between gas concentrations and gas transmittances.

This is now explicitly discussed in the text at the end of section 3.1.3.

Please see manuscript page 6 line 5-16.

(1g) Did you correct for CO2 absorption (it does not appear in Figure 2)?

No, we did not, this has been corrected in the manuscript. In our analysis transmission scalings for NO2 and CO2 are set to 1 (implying no absorbance). This is consistent with the Liang et al, 2005 figure 2.10, which we have referenced in the manuscript.

$NO_2$ and $CO_2$ scaling factors are set to one as their absorption is assumed to be

negligible (see Liang et al., (2005) figure 2.10).

(1h) An explanation as to how and why the approach differs the one documented in Smyth et al. (2004).

This has now been provided in the first paragraph of section 2

Previous, efforts to derive visible channel $R_{rs}$ from the AVHRR catalogue (e.g. Groom and Holligan (1987) and Smyth et al. (2004)) typically use the raw, instrument counts as a starting point to calculate per-channel TOA radiance. In order to apply this approach across the lifetime of a single AVHRR sensor, the radiance must be calibrated according to the sensor degradation parameters. However, as sensor degradation parameters are only available for AVHRR sensors on NOAA-7, 9, 11 and 14 (Rao and Chen , 1995, 1996), the approach is not applicable for analysis of long-term global signals. Consequently, here we adopt a modified version of the approach used by Groom and Holligan (1987), and updated by Smyth et al. (2004), which uses the TOA reflectances as a starting point for the atmospheric correction procedure. The approach is fully detailed in section 3.1.3.

(1g) Did you correct for aerosol transmittance (not mentioned)?

No, aerosol transmittance is not included in the gas transmittance calculations. The aerosol contribution to reflectance is dealt with solely in the balancing of channel-1 and channel-2 reflectances, as performed by Smyth et al, (2004) and used in Uz et al., (2013).

(1h) Is there a sunglint correction?

The PATMOS v5.3 dataset quality flags include a mask for sunglint. This was applied in the initial quality control of the data (discussed in section 3.1.1). No further sun glint correction is applied. This is noted in the text on page 6, line 17

Sunglint is explicitly flagged in, and removed from, the Px5.3 datasets, and no further correction for sunglint is applied.

(1i) Briefly describe the whitecap correction you used.

The whitecap correction is now more explicitly discussed on page 5, line 17-22.

(2) Rrs(VIS) validation:

Even though a qualitative validation is presented in Figure 3 through a visual comparison of four scenes from ocean colour satellites and corresponding AVHRR reflectance products, this work would clearly benefit from a more quantitative assessment. A cross-validation between Rrs(VIS) from appropriate ocean colour satellite bands and Rrs(VIS AVHRR) is probably the best way to validate the dataset produced. Additionally you could compare bloom extent from AVHRR with the coccolithophore bloom mask of Brown and Yoder applied to ocean colour satellites. There is about 20 years of overlapping satellite observations between AVHRR and ocean colour satellites; this should be plenty to make a quantitative assessment of the quality of your dataset. Further, many older papers provided AVHRR satellite data of Emiliania blooms at the original resolution of the sensor, 1km. A comparison with those data would allow you to estimate just how problematic the degradation of the spatial resolution by PATMOS is.

This point is well received, and, in principal the authors agree with the reviewers understandable request for a more thorough validation of the dataset. However, our interpretation of the submission criteria for ESSD (and ESSDD) is that is does not allow for comparisons with other methods and data sets in any fora except for through review articles, which this is not. This limitation has steered our approach to a 'qualitative' evaluation of the data product. We would benefit from editorial input on this point before we proceed to address the concerns raise above in point (2).

(3) Detection limit and uncertainties of the retrieved Rrs and bloom product

Just how reflective does a bloom need to be in order to be picked up by AVHRR satellites and how does the detection limit differ among missions? Can you give typical concentrations of coccolith(ophore)s corresponding to those reflectance detection limits?

These questions are relevant for biologists, ecologists, and biogeochemists interested in coccolithophore blooms. Even if the detection limit would vary across sensors and with illumination and viewing geometry, it would be necessary to document this both in the paper as well as an additional variable in the dataset.

Detection limits for blooms are now provided, based on in situ data taken from Gordon et al., 2001. As expected, it is likely that only the brightest blooms are detected, typically those that are in a decaying phase once liths have been shed and the blooms are at the their brightest. This short analysis is documented in a new "Limits of detection" section. Cross-calibration of the various platforms, and correction for sensor degradation is a key focus of the PATMOS-x processing, and as such it qualifies as a Climate Data Record. Consequently, it does not seem likely that the sensitivity threshold for bloom detection changes between instrument, though we do acknowledge that the change in overpass frequency as more instruments are placed in orbit may have an effect.

Please see the new section, 6.2, which deals with this point.

(4) Another major comment relates to the poor spatial and temporal resolution of the remote sensing reflectance dataset which is monthly at 0.1 x 0.1. As mentioned in the paper this is a spatial coarsening by two orders of magnitude from the original resolution of AVHRR satellite data. Such a coarsening apparently results in a "dimming" of coccolithophore bloom intensity, which is even further exacerbated by making monthly composites. Uz et al. (2014) [Monitoring a Sentinel Species from Satellites: Detecting Emiliania huxleyi in 25 Years of AVHRR Imagery] also used a consistently calibrated dataset of AVHRR data processed with CLAVR-X at 0.25 x 0.25 spatial resolution. Why is this dataset not used instead? It gives much better spatial resolution, and will partly solve your problem of "bloom dimming". Further, as your work is very similar to the work of Uz et al it is an important reference currently missing in your paper. Another (easier) way to mediate the problem of bloom dimming is by increasing the temporal resolution from monthly to weekly. I think weekly resolution data would also allow users

to investigate bloom phenology which is not possible with monthly data.

The reviewer is absolutely correct in their assertion that we have not given proper mention to the work of Uz et al., 2013 in the paper. This point has been addressed throughout. However, there are a number of reasons that the PATMOS-x data is more suited to this analysis. Firstly, the resolution of PATMOS, gridded at 0.1 x 0.1 degrees, is an order of magnitude better than the 0.25 x 0.25 degree grid used by Uz et al., (2014) in their analysis. Secondly, the PATMOS-x record now spans 40 years, allowing for the record to be analysed against climate modes that would not be possible with only 25 year's worth of data. Lastly, unlike CLAVR-X, the PATMOS-x data sets are specifically optimised for climate studies, producing CDR quality records of top of atmosphere reflectances. The authors feel that these advantages justify the choice of PATMOS-x as a start point for producing our data set.

The expected use of the data is for consideration of bloom occurrence within the context of climate modes, and therefore the authors felt that archiving at monthly resolution was both feasible and practical. We also acknowledge that the daily record, will be of use to some researchers (including those concerned with bloom phenology), but remote archiving of a dataset of this size is not feasible. However, as noted in the manuscript, we have made the full daily record available on request.

(5) My last major comment relates to the temporal filtering applied to detect blooms (P5L32-33). I think you are missing out on quite a bit of bloom area by considering as blooms only those observations which have Rrs more than two standard deviations above the corresponding monthly climatological mean. If annual blooms occur in an area in the same month almost every year and last a couple of weeks (which they typically do!) this will give rise to high climatological mean values. Your filtering will miss those blooms as they are considered "background".

In part we agree with the reviewer on this point, however, we are presented with a difficult balancing act with regard to retaining maximum bloom coverage, while reduc-
ing false positives. Although blooms may last for a number of weeks, the sensitivity of AVHRR means that only the brightest part of blooms are detected (lith concentration > 50,000 ml-1). While we acknowledge the repeated occurrence of blooms may result in some filtering of legitimate signal, this phase of blooms does not typically last weeks. We should also clarify that blooms are labelled as such where the Rrs value is 2 standard deviations in the monthly product above the corresponding mean (this text has been updated in the paper).

Please see page 7, lines 23 - 29

**Minor comments:**

(1) P1L9: Rrs detection limit needs to be quantified in the abstract

This has been added to the abstract on page 1, line 9-10

(2) P1L17-18: You should explain why Emiliania huxleyi blooms result in bright waters.

A clarifying statement has been added to the introduction on page 1 lines 19-20

(3) P1L22: A reference to the paper by Winter et al (2014) [Poleward expansion of the coccolithophore Emiliania huxleyi, https://academic.oup.com/plankt/article/36/2/316/1500265 ] would be appropriate.

This reference has been added as requested.

(4) P2L10: "Until now" but see Uz et al.

This section of the introduction has been re-written to include the contributions of Uz et al. 2013, and to better justify the generation and use of the new PATMOS-x based data set.

See page 2 lines 13-18.

(5) P2L28: what about the dataset used in Uz et al?

As above, Uz et al, (2013) is now discussed in the text on page 2.

(6) P2L32: worth mentioning the spectral width of Channel 2.

Channel widths are now specified on page 3 line 10.

(7) P5L14-16: Why don't you just mask high solar zenith angles? As it is now you are loosing good quality clear water, clear atmosphere data.

Masking high solar zenith angles results in substantial loss of coverage, irrespective of in-water content. While we may be losing good clear-water data by masking where signal is very low, it is unlikely that we are losing any blooms through this approach.

(8) P6L11: I would recommend applying a bathymetry mask poleward of 47 as well as false positives are also found on the shallow shelves of the Bering (60N) and Barents (70N)Seas.

We agree with the reviewer, however, implementing a bathymetric filter in this region results in a loss of legitimate signal in the North Sea and English Channel. We rely on the filtering procedure to remove false positives. We cannot conversely retrieve good data from shallow waters.

(9) P8L26: do you have an explanation for this?

We suspect that this is due to some latent contamination from glacial outflows. The text has been updated to reflect this point on page 10 line 16.

The signals along the coast of Greenland and in the Southern Ocean are stronger than anticipated, likely due to some remaining influence of ice and glacial river outflow.

(10) P8L33: reference missing.

(11) P8 Section 5.2: Your maps in Figure 4 are further consistent with other studies documenting poleward expansion of Emiliania huxleyi blooms by Winter et al. (2014) and Neukermans et al. (2018) [Increased intrusion of warming Atlantic water leads to rapid expansion of temperate phytoplankton in the Arctic, https://onlinelibrary.wiley.com/doi/abs/10.1111/gcb.14075 ]. Both references are missing in the paper.

We thank the reviewer for pointing this out, and have added a paragraph to section 5.2 pointing out these consistencies.

(12) P9L3-4: typical is used twice

This has been corrected.

(13) P9L9: "By similar logic" – strange comment to make as blooms of non-calcifying phytoplankton are typically detected based on absorption features, not backscattering.

This point was not well conveyed and has been re-written based on the reviewers suggestion.

Please see page 10 line 30.